# CHA2DS2-VASc Score for Major Adverse Cardiovascular Events Stratification in Patients with Pneumonia with and without Atrial Fibrillation

**DOI:** 10.3390/jcm10184093

**Published:** 2021-09-10

**Authors:** Bo-Yuan Wang, Fei-Yi Lin, Min-Sho Ku, Yu-Hsun Wang, Kun-Yu Lee, Sai-Wai Ho

**Affiliations:** 1Department of Emergency Medicine, Chung Shan Medical University Hospital, Taichung 402, Taiwan; mauspersky@gmail.com (B.-Y.W.); gh20323furby@hotmail.com (K.-Y.L.); 2Department of Emergency Medicine, Chung Shan Medical University, Taichung 402, Taiwan; 3Department of Nursing, Chung Shan Medical University Hospital, Taichung 402, Taiwan; csha891@csh.org.tw; 4Department of Nursing, Chung Shan Medical University, Taichung 402, Taiwan; 5Division of Allergy, Asthma and Rheumatology, Department of Pediatrics, Chung Shan Medical University Hospital, Taichung 402, Taiwan; a129184@yahoo.com.tw; 6School of Medicine, Chung Shan Medical University, Taichung 402, Taiwan; 7Department of Medical Research, Chung Shan Medical University Hospital, Taichung 402, Taiwan; cshe731@csh.org.tw

**Keywords:** CHA2DS2-VASc score, major adverse cardiovascular events, pneumonia

## Abstract

Background: Recent studies have shown an association between CHA2DS2-VASc (congestive heart failure, hypertension, age ≥ 75 years, diabetes mellitus, stroke or transient ischemic attack (TIA), vascular disease, age 65 to 74 years, sex category) score and outcome of acute myocardial infarction, stroke, and chest pain. As pneumonia can affect the cardiovascular system, this study aimed to investigate the performance of the CHA2DS2-VASc score for major adverse cardiovascular events (MACEs) risk stratification in patients with pneumonia. Methods: A retrospective population-based cohort study including 61,843 patients with pneumonia. These patients were divided into two cohorts that were stratified based on the presence or absence of underlying atrial fibrillation (AF). We calculated the CHA2DS2-VASc score and incidence density rates of MACEs in each cohort. Cox regression was conducted to calculate hazard ratio of MACEs in pneumonia patients. The diagnostic performance of CHA2DS2-VASc with regard to MACEs was tested using the receiver operator characteristic curve. Results: Pneumonia patients with higher CHA2DS2-VASc score were more likely develop MACEs in both the AF and non-AF groups. In the AF group, the areas under the curve (AUC), sensitivity, and specificity were 0.824 (0.7773–0.8708), 0.7, and 0.84 respectively. In the non-AF group, the AUC, sensitivity, and specificity were 0.8185 (0.8152–0.8217), 0.75, and 0.83 respectively. Conclusions: The CHA2DS2-VASc score showed good performance in the prediction of MACE in patients with pneumonia.

## 1. Introduction

Pneumonia is the most common infectious disease leading to high mortality and morbidity worldwide [1,2]. Previous studies have shown that pneumonia is associated with an increased risk of cardiovascular complications and short- and long-term mortality [3,4,5,6,7,8,9]. During the acute pneumonia stage, multiple mechanisms have been proposed that could directly and indirectly affect the cardiovascular system, leading to major adverse cardiovascular events (MACEs) including cardiac arrhythmias, acute coronary syndromes, new or worsening heart failure, and stroke in a substantial proportion of patients [10,11]. Therefore, identifying patients with pneumonia at a high risk of MACE and providing strategies for early prevention is of clinical importance in emergency departments. Several risk factors including older age, nursing home residence, preexisting cardiovascular disease, severity of pneumonia, use of insulin, and inpatient care have been reported [3,4,7,12]. To the best of our knowledge, there is no suitable predictive tool in clinical settings for the identification of patients with pneumonia at risk for subsequent MACE.

The CHA2DS2-VASc score is a clinical prediction rule for estimating the risk of stroke in patients with atrial fibrillation (AF) [13]. The components of the CHA2DS2-VASc score include congestive heart failure, hypertension, age ≥ 75 years, diabetes mellitus, prior stroke or transient ischemic attack (TIA) or thromboembolism, vascular disease, age 65–74 years, and sex category. Apart from the prediction of the risk of stroke, recent studies have shown an association between the score and outcome of acute myocardial infarction, stroke, and chest pain [14,15,16]. The CHA2DS2-VASc score is simple to apply in clinical practice. With early recognition and improved ability to stratify high-risk pneumonia-associated cardiac complication patients during admission, it is possible to improve the outcomes of these patients by early prevention, early detection, and early treatment of the events. Therefore, this study aimed to investigate the performance of the CHA2DS2-VASc score for predicting subsequence MACE in patients with pneumonia using a large population-based national database.

## 2. Materials and Methods

### 2.1. Data Source

A retrospective population-based cohort study was conducted using the Taiwan Health and Welfare Data, which are regulated by the Health and Welfare Data Science Center, Ministry of Health and Welfare, Taiwan. The dataset comprises two million Taiwanese residents enrolled in the mandatory National Health Insurance program. It consists of complete information on patients, including the registry for beneficiaries, administrative claims in ambulatory care visits and admissions, diagnosis, medication, and interventions. The disease diagnosis codes in the national health insurance research datasets were derived from the International Classification of Diseases, Ninth Revision, Clinical Modification (ICD-9-CM) codes. This diagnosis coding is highly reliable because all insurance claims were monitored by medical reimbursement specialists and peer reviewers. Our study protocol was approved by the institutional review board of the local hospital (CSMU No: CS1-20033). Written informed consent from the subjects was waived since all data were de-identified.

### 2.2. Study Participants

Patients newly diagnosed with pneumonia (ICD-9-CM codes 481, 482, 483, 485, and 486) between 1 January 2001, and 31 December 2014, with emergency department visits or hospital admission records with the above-mentioned pneumonia ICD-9-CM coding were enrolled in this study. The index date was defined as the date when pneumonia was first diagnosed in the patient. For each patient, the CHA2DS2-VASc score was calculated accordingly. AF was a risk factor for thromboembolic events; hence, the study sample was divided into two cohorts stratified based on the presence or absence of concomitant underlying AF (ICD-9-CM code 427.31): AF and non-AF groups. All patients were followed up until the diagnosis of MACE, death, or until the end of 2015.

MACEs were defined as (1) diagnosis of ischemic stroke (ICD-9-CM codes 433–438) and hemorrhagic stroke (ICD-9-CM codes 430–432); (2) diagnosis of myocardial infarction (ICD-9-CM codes 410–414); (3) diagnosis of heart failure (ICD-9-CM codes 428, 402.01, and 402.91); and (4) death.

Exclusion criteria included patient death on the index day of pneumonia diagnosis and patients aged < 20 years. We also excluded patients newly diagnosed with pneumonia after December 2014 from the analysis because they could not be observed for a long time period (tracked for <1 year). 

Baseline demographic characteristics, such as age and sex, were recorded. For each patient with pneumonia, the CHA2DS2-VASc score was calculated on the index date. One point was assigned for age between 65 and 74 years, history of hypertension (ICD-9-CM codes 401–405), diabetes mellitus (ICD-9-CM code 250), vascular disease (ICD-9-CM codes 410, 411.0, 412, 429.79, 440.0, 440.20–440.24, 440.29, 440.30–440.32, 443.81, 443.89, and 443.9), congestive heart disease (ICD-9-CM code 428), and female sex. Two points were assigned for a history of stroke or TIA (ICD-9-CM codes 433-438) or age ≥ 75 years. Associated comorbidities, including dyslipidemia (ICD-9-CM codes 272.0–272.4), chronic kidney disease (ICD-9-CM code 585), and chronic obstructive pulmonary disease (ICD-9-CM codes 491, 492, and 496) were also documented. Medications included usage of aspirin and other antiplatelet agents (dipyridamole, clopidogrel, ticlopidine, dipyridamole, abciximab, eptifibatide, tirofiban, ticagrelor, cilostazol) were recorded during hospital admission.

### 2.3. Statistical Analysis

Continuous variables were presented as means ± standard deviations and were compared using the independent *t*-test. Categorical data were presented as counts and percentages and were compared using the chi-square test. The incidence density of MACEs in the AF and non-AF groups was calculated by dividing the number of events by person-years at risk. The risk of MACE for patients with each CHA2DS2-VASc score was assessed using Cox regression analysis. The diagnostic performance of CHA2DS2-VASc in predicting MACE was tested using the receiver operator characteristic (ROC) curve, and the area under the curve (AUC) was estimated accordingly. The optimum cutoff point to discriminate MACE was assessed as the maximum sum of sensitivity and specificity using ROC curves. Statistical analyses were performed using SAS software (version 9.4; SAS Institute, Cary, NC, USA). Statistical significance was set at *p* < 0.05.

## 3. Results

Between January 2001 and December 2014, 62,147 patients with pneumonia were included in this study. After exclusions, 1996 patients had pre-existing AF (AF group) and 59,877 did not have pre-existing AF (non-AF group). The study framework is illustrated in Figure 1.

The baseline characteristics, comorbidities, and medication usage of patients with pneumonia in the AF and non-AF groups are shown in Table 1. Compared with the patients with pneumonia in the non-AF group, those in the AF group were significantly older (78.1 ± 10.8, vs. 59.3 ± 20.8), with more comorbidities such as dyslipidemia, chronic kidney disease, chronic obstructive pulmonary disease, and a higher CHA2DS2-VASc score (4.3 ± 1.7 vs. 2.1 ± 1.8). The incidence density of MACE was 2054 (per 1000 person-years) and 170.1 (per 1000 person-years) in the AF and non-AF groups, respectively. Most MACEs developed within 30 days after the diagnosis of pneumonia (Figure 2).

The adjusted hazard ratio (HR) of MACE according to the CHA2DS2-VASc score is shown in Table 2. The adjusted HR of MACE increased from 1.6 (1.68–1.72) to 14.62 (15.96–17.42) when the CHA2DS2-VASc score increased from 0 to ≥6 in the non-AF group. The adjusted HR of MACE increased from 1.72 (1.02–2.92) to 6.59 (3.82–11.35) when the CHA2DS2-VASc score increased from 0 to ≥6 in the AF group.

To investigate the performance of the CHA2DS2-VASc score for predicting major adverse cardiovascular events (MACEs) in patients with pneumonia, the optimum cutoff points of ROC curve, sensitivity (SN), specificity (SP), positive predictive value (PPV), and negative predictive value (NPV) are summarized in Table 3. Figure 3 shows the ROC curves in the prediction of major adverse cardiovascular events in non-AF and AF patients with pneumonia.

## 4. Discussion

In this study, we found that the incidence of MACE increased in line with the CHA2DS2-VASc score in patients with pneumonia with or without AF. Overall, MACE occurred in 60.6% of patients with pneumonia during follow-up. The timing of MACE during pneumonia infection has also been reported. Corrales-Medina et al. showed that most cardiac complications developed within 1 week [3]. Another study showed that the risk of cardiovascular disease was highest during the first year and remained high through a 10-year observation [5]. In this study, we found that most MACEs developed within 30 days after the diagnosis of pneumonia.

The mechanisms of pneumonia-associated cardiovascular complications included hypoxia, proinflammatory status, platelet activation, thrombosis formation, unstable plaque, increased sympathetic tone, and several adverse effects of antibiotics [11,17,18,19]. As most of the MACEs were due to thromboembolic events, we investigated the CHA2DS2-VASc score to predict MACE. In this study, the CHA2DS2-VASc score showed good calibration and performance in the prediction of MACE during follow-up. With a cutoff value of 2, the SN, SP, PPV, and NPV were 0.76, 0.83, 0.87, and 0.69, respectively, for all patients with pneumonia. Using a cutoff value of 4 for patients with AF, the performance was better than that of patients without AF.

When focusing on ischemic events, the performance was better in predicting acute ischemic stroke than ischemic heart disease. However, when compared with patients without AF, the AUC was smaller for patients with AF in both groups. A significantly higher ratio of antiplatelet agent usage in the AF group may explain this condition. Previous study found that ischemic stroke or transient ischemic attack patients allocated to long-term aspirin therapy had reduced risk of recurrent stroke, myocardial infarction, and vascular death by 13% [20].

A previous study showed that approximately 10–15% of patients developed heart failure after pneumonia infection [6,17]. Although the mechanism is unclear, several hypotheses have been proposed, including persistent inflammatory state, hypoxia, tachycardia, specific microbacteria, and fluid overload during the management of pneumonia infection [4,10,21,22]. In this study, the incidence of heart failure was lower than that of ischemic stroke and ischemic heart disease. With a cutoff value of 2, the SN, SP, PPV, and NPV were 0.85, 0.55, 0.32, and 0.94, respectively. A very high NPV for the occurrence of acute heart failure suggests the utilization of the CHA2DS2-VASc score for the selection of low-risk patients. 

Several studies have documented an increase in the risk of MACE within the first 30 days after acute respiratory infection. Violi et al. demonstrated that cardiovascular complications were associated with a five-fold increase in pneumonia-associated 30-day mortality [7]. Corrales-Medina et al. also showed a 60% increase in the risk of death at 30 days in patients with pneumonia who developed cardiac complications [3]. Although the risk of cardiovascular complications was the highest in the first 30 days after pneumonia, it remained 1.5-fold higher in subsequent years [5]. However, the long-term mortality rate has seldom been investigated. Our study showed a long-term mortality rate of 39% during the follow-up. The performance of the CHA2DS2-VASc score in the prediction of all-cause mortality was also good. With a cutoff value of 2, the SN, SP, PPV, and NPV were 0.84, 0.67, 0.62, and 0.86, respectively.

This study had some limitations. First, the database was primarily used for insurance reimbursement; therefore, it did not contain any physiological and laboratory data of the studied patients. We did not know the severity of the disease, such as C-reactive protein level, the pneumonia severity index, CURB-65, Modified Early Warning Score, and sequential organ failure assessment, which are associated with MACE occurrence and mortality [3,7,8,17,21,22]. Moreover, the severity of underlying heart failure such as ejection fraction of left and right ventricle of the patient could not be determined from the database. Patients with higher severity of heart failure have a higher rate of MACE. Second, risk factors for MACE, such as cigarette smoking and obesity, cannot be obtained from the claims database. Third, we could not determine the patients’ medication compliance. Noncompliance with antiplatelet and anticoagulant medications, especially in patients with AF, increased the risk of MACE. Fourth, in Taiwan, oral anticoagulants such as dabigatran, rivaroxaban, apixaban, and edoxaban have been approved by the Taiwan Food and Drug Administration since June 2014. Therefore, the study database did not contain anticoagulant treatment of the patients. The absence of anticoagulant treatment for AF patients with CHA2DS2-VASc more than 2 have a higher rate of ischemic stroke. Fifth, the pneumonia patients in this study did not include COVID-19 patients. Recent studies found that CHA2DS2-VASc score is useful to predict in-hospital mortality but does not predict embolic risk in COVID-19 patients. Therefore, the results should be read and used with caution in the pandemic situation [23,24]. 

## 5. Conclusions

In conclusion, this is the first study that investigated the clinical CHA2DS2-VASc score to predict MACE in patients with pneumonia. For patients with pneumonia, the incidence of MACE increased in line with the CHA2DS2-VASc score. The CHA2DS2-VASc score showed good performance in the prediction of MACE and all-cause mortality. 

## Figures and Tables

**Figure 1 jcm-10-04093-f001:**
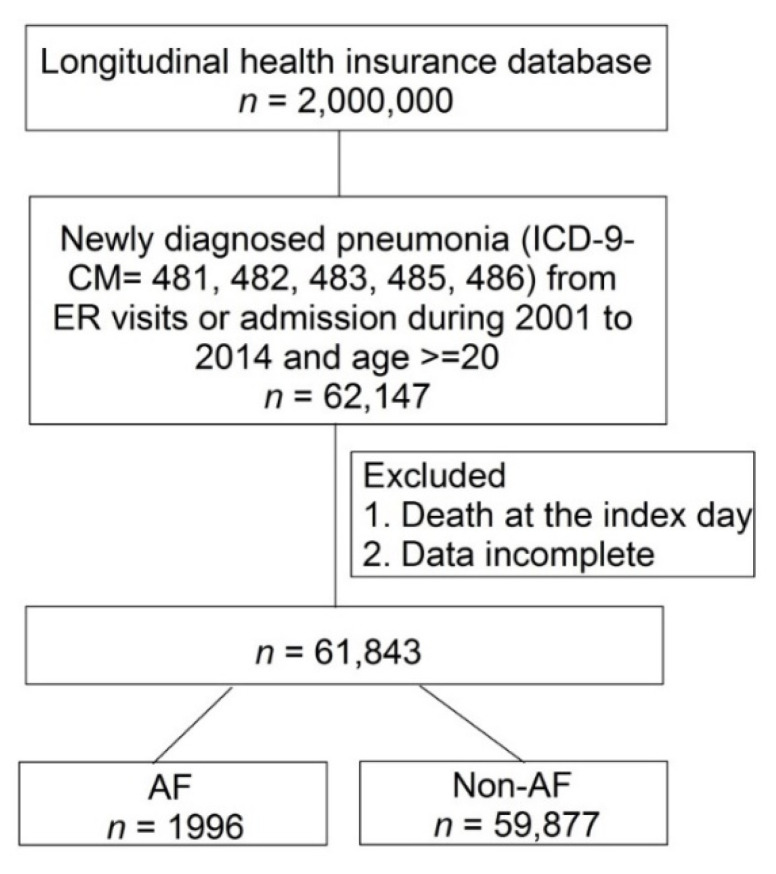
Flow chart of patient selection.

**Figure 2 jcm-10-04093-f002:**
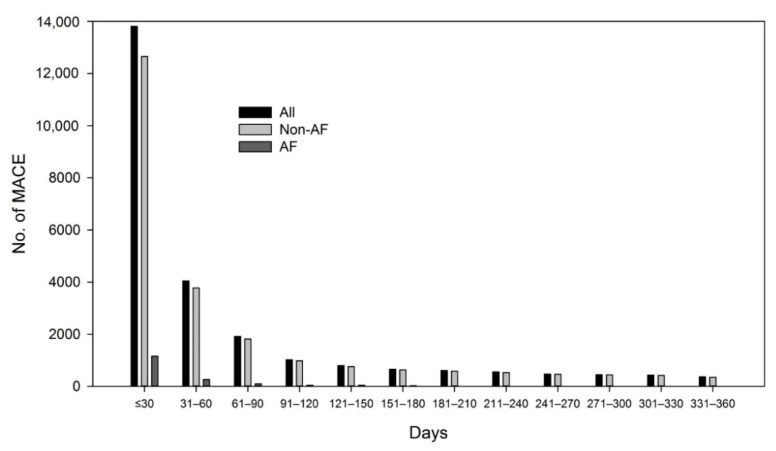
Time elapsed between pneumonia and development of major adverse cardiovascular events (MACE). AF, atrial fibrillation.

**Figure 3 jcm-10-04093-f003:**
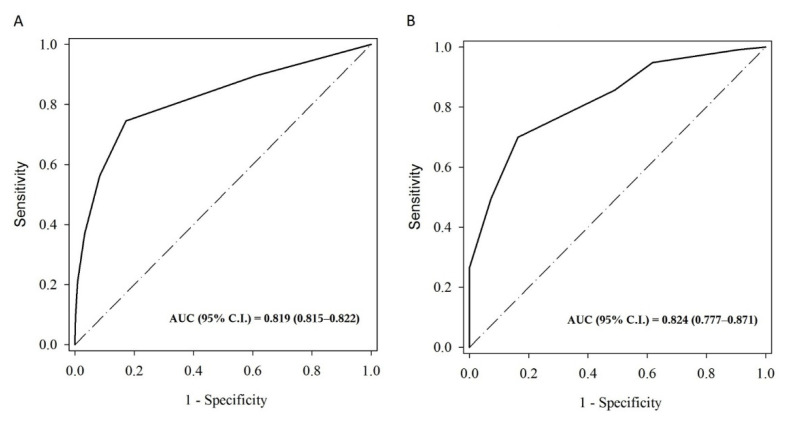
ROC curves in the prediction of major adverse cardiovascular events in non-AF (**A**) and AF (**B**) patients with pneumonia. AUC, areas under the curve; AF, atrial fibrillation; ROC, receiver operator characteristic.

**Table 1 jcm-10-04093-t001:** Demographic characteristics of pneumonia patients with and without atrial fibrillation.

	All (*n* = 61873)	Non-Atrial Fibrillation (*n* = 59877)	Atrial Fibrillation (*n* = 1996)	*p* Value
Age				<0.001
20–39	13,471 (21.8)	13,460 (22.5)	11 (0.6)	
40–64	19,422 (31.4)	19,216 (32.1)	206 (10.5)	
≥65	28,950 (46.8)	27,201 (45.4)	1749 (89.0)	
Mean ± SD	59.9 ± 20.8	59.3 ± 20.8	78.1 ± 10.8	<0.001
Sex				0.025
Female	26,882 (43.4)	26,076 (43.5)	806 (41.0)	
Male	34,961 (56.5)	33,801 (56.5)	1160 (59.0)	
CHA2DS2-VASc				
Congestive heart failure	3921 (6.3)	3094 (5.2)	827 (42.1)	<0.0001
Hypertension	23,116 (37.4)	21,742 (36.3)	1374 (69.9)	<0.0001
Diabetes	12,223 (19.8)	11,627 (19.4)	596 (30.3)	<0.0001
Vascular disease	1683 (2.7)	1541 (2.6)	142 (7.2)	<0.0001
Prior ischemic stroke/transient ischemic attack	8851 (14.3)	8028 (13.4)	823 (41.9)	<0.0001
Mean ± SD	2.2 ± 1.8	2.1 ± 1.8	4.3 ± 1.7	<0.0001
Other comorbidities				
Hyperlipidemia	7404 (12.0)	7093 (11.8)	311 (15.8)	<0.0001
Chronic kidney disease	3173 (5.1)	2892 (4.8)	191 (9.7)	<0.0001
Chronic obstructive pulmonary disease	5483 (8.9)	5035 (8.4)	448 (22.8)	<0.0001
Antiplatelet agents	13,716 (22.2)	12,977 (21.7)	739 (37.6)	<0.0001
Aspirin	21,414 (34.6)	20,670 (34.5)	744 (37.8)	0.002
Follow-up durations (years), Mean ± SD	3.4 ± 4.0	3.5 ± 4.0	0.5 ± 1.3	<0.0001

Abbreviations: SD, standard deviations; CHA2DS2-VASc, congestive heart failure, hypertension, age ≥ 75 years, diabetes mellitus, stroke or transient ischemic attack (TIA), vascular disease, age 65 to 74 years, sex category.

**Table 2 jcm-10-04093-t002:** Cox proportional hazard model analysis for risk of MACE.

			Univariate		Multivariate ^†^	
	*n*	No. of MACE	HR (95% C.I.)	*p* Value	HR (95% C.I.)	*p* Value
Non-AF						
CHA2DS2-VASc						
0	13,167	3706	Reference		Reference	
1	16,032	5357	1.25 (1.36–1.31)	<0.0001	1.60 (1.76–1.68)	<0.0001
2	8672	6517	4.58 (4.97–4.77)	<0.0001	2.96 (3.30–3.12)	<0.0001
3	8103	6865	6.69 (7.26–6.97)	<0.0001	4.17 (4.68–4.42)	<0.0001
4	6279	5697	9.27 (10.10–9.68)	<0.0001	5.90 (6.68–6.28)	<0.0001
5	4262	4102	13.74 (15.07–14.39)	<0.0001	8.64 (9.85–9.23)	<0.0001
6	2320	2281	18.00 (20.07–19.01)	<0.0001	11.14 (12.85–11.97)	<0.0001
>6	1042	1032	19.91 (22.94–21.37)	<0.0001	14.62 (17.42–15.96)	<0.0001
AF						
CHA2DS2-VASc						
0	22	17	Reference		Reference	
1	98	82	1.34 (0.79–2.25)	0.277	1.72 (1.02–2.92)	0.044
2	182	175	1.61 (0.98–2.65)	0.061	2.27 (1.35–3.81)	0.002
3	318	300	1.93 (1.18–3.15)	0.009	2.95 (1.75–4.97)	<0.0001
4	396	391	2.85 (1.75–4.64)	<0.0001	4.30 (2.54–7.26)	<0.0001
5	443	439	3.24 (1.99–5.28)	<0.0001	5.08 (3.00–8.61)	<0.0001
6	286	286	3.72 (2.27–6.09)	<0.0001	5.91 (3.46–10.10)	<0.0001
>6	221	221	3.91 (2.38–6.43)	<0.0001	6.59 (3.82–11.35)	<0.0001

Adjusted for age, sex, hyperlipidemia, chronic kidney disease, chronic obstructive pulmonary disease, usage of antiplatelet agents and aspirin. ^†^ Abbreviations: AF, atrial fibrillation; HR, hazard ratio; MACE, major adverse cardiovascular events.

**Table 3 jcm-10-04093-t003:** Diagnostic performance of CHA2DS2-VASc score in predicting of MACE.

	*n*	No. of Event	AUC (95% C.I.)	Cut-Off	Sensitivity	Specificity	PPV	NPV
All event								
All	61,843	37,468	0.8247 (0.8215–0.8278)	2.0	0.7555	0.8270	0.8703	0.68751
NonAF	59,877	35,557	0.8185 (0.8152–0.8217)	2.0	0.7451	0.8280	0.8636	0.68961
AF	1966	1911	0.8240 (0.7773–0.8708)	4.0	0.6996	0.8364	0.9933	0.07419
Ischemic heart disease								
All	61,843	16,658	0.6588 (0.6542–0.6634)	2.0	0.7391	0.5527	0.3786	0.85177
NonAF	59,877	15,810	0.6602 (0.6555–0.6649)	2.0	0.7283	0.5651	0.3753	0.85287
AF	1966	848	0.5399 (0.5146–0.5653)	4.0	0.5554	0.5125	0.4071	0.51613
Heart failure								
All	61,843	12,180	0.7424 (0.7379–0.7470)	2.0	0.8489	0.5533	0.3179	0.93724
NonAF	59,877	11,064	0.7379 (0.7332–0.7427)	2.0	0.8382	0.5615	0.3023	0.93870
AF	1966	1116	0.5480 (0.5227–0.5733)	5.0	0.5117	0.5541	0.6011	0.46358
Acute ischemic stroke								
All	61,843	15,463	0.7720 (0.7678–0.7763)	3.0	0.6957	0.7216	0.4545	0.87672
NonAF	59,877	14,543	0.7710 (0.7667–0.7754)	2.0	0.8296	0.5894	0.3933	0.91513
AF	1966	920	0.6731 (0.6498–0.6964)	5.0	0.6261	0.6425	0.6063	0.66142
Intracranial hemorrhage								
All	61,843	2678	0.6488 (0.6387–0.6589)	2.0	0.7737	0.4853	0.0637	0.97933
NonAF	59,877	2559	0.6514 (0.6410–0.6617)	2.0	0.7644	0.4989	0.0638	0.97935
AF	1966	119	0.5343 (0.4821–0.5866)	5.0	0.5378	0.5203	0.0674	0.94587
Death								
All	61,843	24,182	0.7938 (0.7902–0.7974)	2.0	0.8358	0.6731	0.6214	0.86459
NonAF	59,877	22,684	0.7899 (0.7862–0.7936)	2.0	0.8271	0.6796	0.6115	0.86565
AF	1966	1498	0.6854 (0.6576–0.7131)	4.0	0.7517	0.5299	0.8366	0.40000

Abbreviations: AF, atrial fibrillation; AUR, areas under the curve; MACE, major adverse cardiovascular events; NPV, negative predictive value; PPV, positive predictive value; AUC, areas under the curve.

## Data Availability

Data are available at National Health Insurance Research Database, Taiwan (http://nhird.nhri.org.tw/en/index.html, accessed on 5 July 2011). The data utilized in this study cannot be made available due to the “Personal Information Protection Act” executed by Taiwan’s government, starting from 2016. Requests for data can be sent as a formal proposal to the NHIRD_MOHW (http://nhird.nhri.org.tw, accessed on 31 August 2020) or by email to nhird@nhri.org.tw.

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
