# Peer review of "CHA2DS2-VASc Score for Major Adverse Cardiovascular Events Stratification in Patients with Pneumonia with and without Atrial Fibrillation"

_jcm, 2021, doi:10.3390/jcm10184093_

Round 1

Reviewer 1 Report

The paper aims to evaluate the accuracy of the CHA2DS2-VASc score to predict MACE after a discharge diagnosis of pneumonia from ER or admission at Taiwan (2001-2014). Multicenter cohort study. 

Major comments
There is no mention on the anticoagulant treatment of patients. The last AF Guidelines specify the need of oral anticoagulants in patients with AF and CHA2DS2-VASc >=2. It needs to be specified in the Table 1. AF patients have a higher MACE rate due to the absence of anticoagulant treatment? The absence of anticoagulant treatment in patients with AF should be entered in the adjustment for the Cox model. 

Other comments
1. Introduction. Well and concise. 
2. Materials and methods. Need to specify which kinds of antiplatelet agents. Were patients previouly on antiplatelet , were these treatment continuated during admission or antiplatelets were started after admission?
3. Results. Well structured. Please, add the whole group data on Table 1, Figure 2 and Table 3. Misspell on Legend of Table 3 "usage of antiplatelet agenets". Consider to add in Table 1 in other comorbidities "Prior Myocardial infarction". Consider to add a score of severity of pneumonia such as CURB-65 or MEWS or any other score that grades pneumonia severity. It may add a bias in the whole study. 
4. Limitations. Please consider to add comments on LV/RV EF. Consider to comment on the use of the score in the actual pandemic situation. 

Author Response

Dear reviewer 1#

We thank the opportunity to improve our manuscript and have attempted to incorporate the suggestions into our revision. The changes within the revised manuscript have been highlighted by using red text.

Comments and Suggestions for Authors

The paper aims to evaluate the accuracy of the CHA2DS2-VASc score to predict MACE after a discharge diagnosis of pneumonia from ER or admission at Taiwan (2001-2014). Multicenter cohort study.

Major comments

There is no mention on the anticoagulant treatment of patients. The last AF Guidelines specify the need of oral anticoagulants in patients with AF and CHA2DS2-VASc >=2. It needs to be specified in the Table 1. AF patients have a higher MACE rate due to the absence of anticoagulant treatment? The absence of anticoagulant treatment in patients with AF should be entered in the adjustment for the Cox model.

Response:

▓In Taiwan, oral anticoagulants such as dabigatran, rivaroxaban, apixaban, and edoxaban have been approved by Taiwan FDA since June 2014. Therefore, the study database did not contain anticoagulant treatment of the patients.

In the limitation section of the revise manuscript, we added “Forth, in Taiwan, oral anticoagulants such as dabigatran, rivaroxaban, apixaban, and edoxaban have been approved by the Taiwan Food and Drug Administration since June 2014. Therefore, the study database did not contain anticoagulant treatment of the patients. The absence of anticoagulant treatment for AF patients with CHA2DS2-VASc more than 2 have a higher rate of ischemic stroke.” [line 267-271]

Other comments

  1. Well and concise.

Response:

▓Thank you for your comments.

  1. Materials and methods. Need to specify which kinds of antiplatelet agents. Were patients previouly on antiplatelet , were these treatment continuated during admission or antiplatelets were started after admission?

Response:

▓Thank you for your suggestions. In the materials and methods section, we specify antiplatelet agents and changed the sentence “Medications included usage of antiplatelet agents and aspirin were recorded during study period.” To “Medications included usage of aspirin and other antiplatelet agents (dipyridamole, clopidogrel, ticlopidine, dipyridamole, abciximab, eptifibatide, tirofiban, ticagrelor, cilostazol) were recorded during hospital admission.” [line 105-108]

▓Patients were classified in antiplatelet agents usage group if antiplatelet agents were recorded during the admission period. Therefore, we changed the sentence “Medications included usage of antiplatelet agents and aspirin were recorded during study period.” To “Medications included usage of aspirin and other antiplatelet agents (dipyridamole, clopidogrel, ticlopidine, dipyridamole, abciximab, eptifibatide, tirofiban, ticagrelor, cilostazol) were recorded during hospital admission.” [line 105-108]

▓Recently due to the COVID-19 level two alert in Taiwan, we cannot enter the Health and Welfare Data Science Center to export the data to perform the supplementary data to analysis the usage of antiplatelet agents before and after the admission. We hope that these replies may meet your requirement for being published. Thank you very much for your kind assistance.

  1. Well structured. Please, add the whole group data on Table 1, Figure 2 and Table 3. Misspell on Legend of Table 3 "usage of antiplatelet agenets". Consider to add in Table 1 in other comorbidities "Prior Myocardial infarction". Consider to add a score of severity of pneumonia such as CURB-65 or MEWS or any other score that grades pneumonia severity. It may add a bias in the whole study.

Response:

▓Thank you for your comments to improve our manuscript. We added whole group data on Table 1, Figure 2 and Table 3 in the revised manuscript.

▓In legend of table 2, we corrected the sentence “usage of antiplatelet agenets” to “usage of antiplatelet agents” [line 182]

▓We agree that prior Myocardial infarction may be a risk factor for MACEs. However, due to the COVID-19 level two alert in Taiwan, we cannot enter the Health and Welfare Data Science Center to export the data to perform the supplementary data to add prior Myocardial infarction in table 1. We hope that these replies may meet your requirement for being published. Thank you very much for your kind assistance.

▓Thank you for your suggestion. Because the database did not contain the laboratory results, we can not analysis the level of CRP of the study patients. In the limitation section, we did mention that “First, the database was primarily used for insurance reimbursement; therefore, it did not contain any physiological and laboratory data of the studied patients.” In the revised manuscript limitation section, we changed the sentence “We did not know the severity of the disease, such as the pneumonia severity index and sequential organ failure assessment, which are associated with MACE occurrence and mortality” to “We did not know the severity of the disease, such as C-reactive protein level, the pneumonia severity index, CURB-65, Modified Early Warning Score, and sequential organ failure assessment, which are associated with MACE occurrence and mortality” [line 258-261]

  1. Limitations. Please consider to add comments on LV/RV EF. Consider to comment on the use of the score in the actual pandemic situation.

Response:

▓Thanks for your suggestion and to improve this aspect. In the limitation section, we added “Moreover, the severity of underlying heart failure such as ejection fraction of left and right ventricle of the patient could not be determined from the database. Patients with higher severity of heart failure have a higher rate of MACE.” [line 261-263]

In the limitation section, we added “Fifth, the pneumonia patients in this study did not include COVID-19 patients. Recent studies found that CHA2DS2-VASc score is useful to predict in-hospital mortality but does not predict embolic risk in COVID-19 patients. Therefore, the results should be read and used with caution in the pandemic situation.” [272-275]

We also added 2 references for this limitation.

[23] Quisi A, Alıcı G, Harbalıoğlu H, Genç Ö, Er F, Allahverdiyev S, Yıldırım A, et al. The CHA2DS2-VASc score and in-hospital mortality in patients with COVID-19: A multicenter retrospective cohort study. Turk Kardiyol Dern Ars. 2020;48:656-663. English. doi: 10.5543/tkda.2020.03488.

[24] Uribarri A, Núñez-Gil IJ, Aparisi Á, Arroyo-Espliguero R, Maroun Eid C, Romero R, et al.; HOPE COVID-19 investigators. Atrial fibrillation in patients with COVID-19. Usefulness of the CHA2DS2-VASc score: an analysis of the international HOPE COVID-19 registry. Rev Esp Cardiol (Engl Ed). 2021;74:608-615. doi: 10.1016/j.rec.2020.12.009.

Reviewer 2 Report

This is a population-based study investigating the performance of CHA2DS2-Vasc score for MACEs prediction in patients with pneumonia. The diagnostic performance of the score was tested using the ROC curve and the AUC. The optimum cutoff point with the corresponding sensitivity, specificity, PPV and NPV are also presented for the combined endpoint (MACEs) and for individual outcomes (e.g. heart failure, stroke etc). The authors conclude that the incidence of MACE increased in line with the score.

Overall, the paper is well-written. 

Introduction: The authors should clearly mention why they chose to investigate the prognostic uitility of CHA2DS2-Vasc score in patients with pneumonia.

Methods: Well-written and comprehensive. 

Results: My major concern is that the authors did not adjust for important laboratory data (e.g. CRP) associated with the severity of pneumonia. This is a major limitation

Discussion: In lines 207-210 the authors state: "When focusing on ischemic events, the performance was better in predicting acute ischemic stroke than ischemic heart disease. However, when compared with patients without AF, the AUC was smaller for patients with AF in both groups. A significantly higher ratio of antiplatelet agent usage in the AF group may explain this condition". However, in AF is indicated anticoagulation and not antiplatelets. Please explain.

Conslusion: "The performane is slightly better for AF patients". This statement seems a bit arbitrary to me, according to the study's results. Please explain

Author Response

Dear reviewer 2#

We thank the opportunity to improve our manuscript and have attempted to incorporate the suggestions into our revision. The changes within the revised manuscript have been highlighted by using red text.

Comments and Suggestions for Authors

This is a population-based study investigating the performance of CHA2DS2-Vasc score for MACEs prediction in patients with pneumonia. The diagnostic performance of the score was tested using the ROC curve and the AUC. The optimum cutoff point with the corresponding sensitivity, specificity, PPV and NPV are also presented for the combined endpoint (MACEs) and for individual outcomes (e.g. heart failure, stroke etc). The authors conclude that the incidence of MACE increased in line with the score.

Overall, the paper is well-written.

  1. Introduction: The authors should clearly mention why they chose to investigate the prognostic uitility of CHA2DS2-Vasc score in patients with pneumonia.

Response:

▓Thank you for your comments to improve our manuscript. In the introduction section, we added “CHA2DS2-VASc score is simple to apply in clinical practice. By early recognition and improve the ability to stratify high-risk pneumonia-associated cardiac complications patients during admission, it is possible to improve the outcomes of these patient by early prevention, early detection and early treatment of the events.” [line 54-58]

  1. Methods: Well-written and comprehensive.

Response:

▓Thank you for your comments.

  1. Results: My major concern is that the authors did not adjust for important laboratory data (e.g. CRP) associated with the severity of pneumonia. This is a major limitation.

Response:

▓Thank you for your suggestion. In the limitation section, we changed the sentence “We did not know the severity of the disease, such as the pneumonia severity index and sequential organ failure assessment, which are associated with MACE occurrence and mortality” to “We did not know the severity of the disease, such as C-reactive protein level, the pneumonia severity index, CURB-65, Modified Early Warning Score, and sequential organ failure assessment, which are associated with MACE occurrence and mortality” [line 258-261]

We also cited 2 references for this sentence.

[21] Hedlund J, Hansson LO. Procalcitonin and C-reactive protein levels in community-acquired pneumonia: correlation with etiology and prognosis. Infection. 2000;28:68-73. doi: 10.1007/s150100050049.

[22] Ridker PM. Clinical application of C-reactive protein for cardiovascular disease detection and prevention. Circulation. 2003;107:363-369. doi: 10.1161/01.cir.0000053730.47739.3c.

  1. Discussion: In lines 207-210 the authors state: "When focusing on ischemic events, the performance was better in predicting acute ischemic stroke than ischemic heart disease. However, when compared with patients without AF, the AUC was smaller for patients with AF in both groups. A significantly higher ratio of antiplatelet agent usage in the AF group may explain this condition". However, in AF is indicated anticoagulation and not antiplatelets. Please explain.

Response:

▓Thank you for your comments to improve our manuscript. In the revised manuscript, we further explain and added “Previous study found that ischemic stroke or transient ischemic attack patient allocated to long-term aspirin therapy reduced risk of recurrent stroke, myocardial infarction and vascular death by 13%.” [line 232-235]

We also cited a reference for the explanation:

[20] O'Donnell MJ, Hankey GJ, Eikelboom JW. Antiplatelet therapy for secondary prevention of noncardioembolic ischemic stroke: a critical review. Stroke. 2008;39:1638-1646. doi: 10.1161/STROKEAHA.107.497271.

  1. Conslusion: "The performance is slightly better for AF patients". This statement seems a bit arbitrary to me, according to the study's results. Please explain

Response:

▓Thanks for your comment to improve this aspect. The AUC for AF (0.824) is slightly larger than that of Non AF group (0.819). However, we did not test any statistical significance between these two AUC. Therefore, we deleted the sentence “The performance is slightly better for AF patients” in the conclusion section of the revised manuscript.

Round 2

Reviewer 1 Report

Dear authors, 

the major concern is about that there is no mention on how patients with AF were treated. Were they under oral anticoagulants? If not, AF patients may have increased MACE risk. Please, state as previously asked if patients with AF were adequately treated with OAC (vitamin K or non-vitamin K). If not treated, patients with AF have increased worse prognosis because increased MACE. If you do not know how AF patients were treated, the AF sub-analysis should not be included.  Thank you.  

Author Response

Thank you for your comments to improve our manuscript. The study database only contained patients data between January 1, 2001 and December 31, 2014. In this study period, the usage od NOAC was not approval by Taiwan FDA. Therefore, in the limitation section of the revise manuscript, we added

“Forth, in Taiwan, oral anticoagulants such as dabigatran, rivaroxaban, apixaban, and edoxaban have been approved by the Taiwan Food and Drug Administration since June 2014. Therefore, the study database did not contain anticoagulant treatment of the patients. The absence of anticoagulant treatment for AF patients with CHA2DS2-VASc more than 2 have a higher rate of ischemic stroke.”

Due to the COVID-19 level two alert in Taiwan, we cannot enter the Health and Welfare Data Science Center to export the data if AF were treated such as warfarin. Therefore, the study can not in-time provide the supplementary analysis of anticoagulant treatment in AF patients. We hope that these replies may meet your requirement for being published. Thank you very much for your kind assistance.

Reviewer 2 Report

My comments have been addressed

Author Response

Thank you for your great help.